# Transient evolution of permeability and friction in a slowly slipping fault activated by fluid pressurization

Frédéric Cappa [1,2✉], Yves Guglielmi[3] & Louis De Barros [1]

The mechanisms of permeability and friction evolution in a natural fault are investigated in situ. During three fluid injection experiments at different places in a fault zone, we measured simultaneously the fluid pressure, fault displacements and seismic activity. Changes in fault permeability and friction are then estimated concurrently. Results show that fault permeability increases up to 1.58 order of magnitude as a result of reducing effective normal stress and cumulative dilatant slip, and 19-to-60.8% of the enhancement occurs without seismic emissions. When modeling the fault displacement, we found that a rate-and-state friction and a permeability dependent on both slip and slip velocity together reasonably fit the fault-parallel and fault-normal displacements. This leads to the conclusion that the transient evolution of fault permeability and friction caused by a pressure perturbation exerts a potentially dominant control on fault stability during fluid flow.

[1] Université Côte d'Azur, CNRS, Observatoire de la Côte d'Azur, IRD, Géoazur, Sophia Antipolis, France. [2] Institut Universitaire de France, Paris, France. [3] Energy Geosciences Division, Lawrence Berkeley National Laboratory, Berkeley, CA, USA. ✉email: frederic.cappa@univ-cotedazur.fr

Fault friction and permeability are central to understand the mode of fault slip, including seismic and aseismic response, and fluid flow, with major interest to subsurface reservoirs engineering and earthquake hazard mitigation[1,2]. Friction and permeability of faults are commonly measured in the laboratory using rock mechanics experiments on samples with sizes ranging from centimeters to a meter[3–6]. These laboratory-sized experiments are very useful to derive constitutive laws describing the evolution of friction and fluid flow along faults. The friction laws (i.e. rate-and-state or slip-weakening friction) are then used in physics-based models to explain the spectrum of slip behavior on faults[7], while permeability laws (i.e. stress- or strain-dependent permeability) are used to model fluid pressure diffusion[8,9].

Although laboratory-scale experiments are important for precise examination of the mechanisms in controlled environments, questions remain on the extrapolation of laboratory-derived frictional and hydraulic properties to faults at large scale in natural conditions[10]. Part of the difficulty is that simultaneous measurements of fluid pressure and deformation into real-world faults at depth are rare, and consequently, hydromechanical processes and properties are not well constrained. However, recent in situ experiments of fault activation caused by fluid injection at shallow depths (from 0.3 to 1.5 km) have proved to be well adapted for studying at the decametric scale how a fault responds to a known fluid pressure perturbation[11–15]. Such experiments provide quantitative information on the hydromechanical processes[16–18], and significant insights have been gained into the physics of fault slip[11,19]. Yet, to date, constraints on the evolution of permeability and friction during fluid pressurization in a natural fault are rare.

Here, we explore the hydromechanical behavior of a fault zone in response to controlled fluid injection combining the analysis of simultaneous and continuous measurements of fluid pressure and fault displacements in the injection borehole and seismicity in the rock volume that surrounds it. We report on three injection tests that activated the fault in three different zones. The data are used to infer the evolution of fault permeability and friction during increasing fluid pressure, and its coupling with slip.

## Results

**Experimental procedure and fluid injection in a natural fault.**
The experiments were conducted at 280 m depth within the Low Noise Underground Laboratory (France, https://lsbb.cnrs.fr) in a horizontal gallery intersecting a seismically-inactive kilometer-scale fault zone in limestone rocks representative of sedimentary reservoirs (Fig. 1 and Supplementary Fig. 1). The fault zone was mapped on the gallery walls and from the core samples collected on ~20 m long vertical boreholes. The fault zone is 20-m thick and has an average orientation with a dip direction of 30 °N and a dip angle of 70°[11,20]. Based on geological data, the fault zone is made up of a main fault core and multiple principal slip zones embedded in a damaged rock volume with subparallel fractures of 1–10 m length[20] (Fig. 1B). The fracture density progressively decreases away from the fault core. The strike-slip to normal cumulated slip offset is of few meters at the gallery outcrop. Most of the deformation is located in the fault core. The fault planes in the main slip zones are smooth and marked by a thin clay gouge (from few millimeters up to 3 cm thick), while in the damage zone, secondary faults and fractures are rough and without clay content[20]. The limestone layer hosting the fault zone contains bedding planes with a dip direction of 135°N and a dip angle of 25°[20].

The fault zone is initially dry in the unsaturated zone of the carbonate reservoir. Then, pressurized water was injected directly in the tested geological structures using an engine pump. The

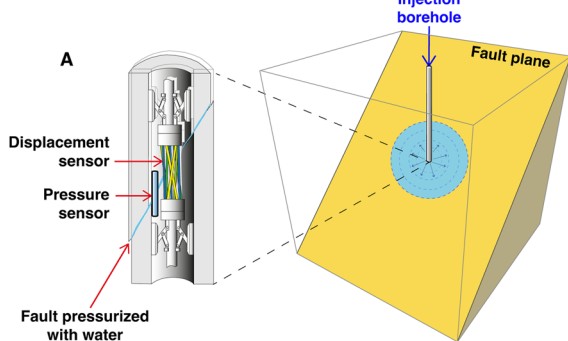

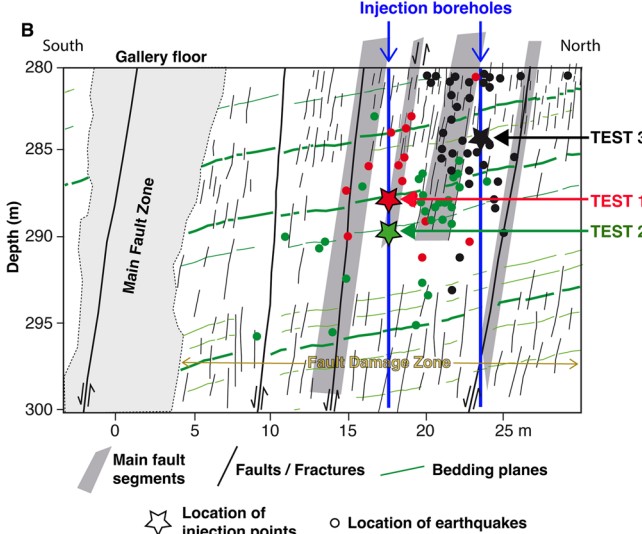

**Fig. 1 Experimental context. A** Set-up for a typical fluid injection test that measures fluid pressure and displacement in the fault plane with a SIMFIP probe[22] in borehole. **B** Locations of the injection tests (red star for Test 1, green star for Test 2, and black star for Test 3) and seismic events (colored dots, different for each test) in the fault zone at 280 m depth within the Low Noise Underground Laboratory in France (modified from[21]). The gallery floor corresponds to the top of the cross-section.

injection procedure corresponds to a pressure-control mode in which the fluid pressure is increased step-by-step into a sealed borehole interval (i.e. injection chamber). Three injection tests were performed at different places in the damage zone on selected secondary fault segments in order to explore the variety of fault hydromechanical response. Test 1 was performed 7.5 m below the gallery floor, 9.5 m for Test 2, and 3.75 m for Test 3 (Fig. 1B; see Tests 2, 3 and 11, respectively, in[21] for further details). Inside the injection chamber, the pressurized fault plane is located at the middle of the interval between two inflatable packers. Tests 1 and 2 are situated along a secondary minor fault plane, while Test 3 is in a fractured zone. In Test 1, the rock matrix around the pressurized fault plane contains three subparallel bedding planes in the sealed borehole interval, while in Test 2, the rock matrix contains micro-fractures without bedding planes. In Test 3, the pressurized fault plane is surrounded by a fractured matrix with a fracture density higher than in Tests 1 and 2. In the three tests, the selected planes are not filled with clay material. The pressurized fault segments have similar dip angles (65° in Test 1, 74° in Test 2, and 80° in Test 3). Among the three tests, the injection duration varies from 1400 s (Test 1), 460 s (Test 2) to 1000 s (Test 3), and fluid pressure is increased from near zero to maximum values ranging from 4.8 (Test 1), 5.02 (Test 2) to

5.88 MPa (Test 3). For each test, the injected fluid volume is 3.8, 0.52, 0.33 m³, respectively. The evolution of fluid pressure, fault-normal (opening) and fault-parallel (slip) displacements were recorded at the injection point with a specially designed hydromechanical probe "SIMFIP" based on high-precision fiber-optic sensors (1 µm for displacement, and 1 kPa for pressure) in a 2.4 m long packer-isolated borehole interval[22] (Fig. 1A).

The seismic activity is monitored at proximal distances of the injection with a network of 22 accelerometers and 9 geophones located both in boreholes and at the gallery floor. This seismic network configuration (Supplementary Fig. 1b) allows to locate micro-seismicity with a precision of 1.5 m, and to measure moment magnitudes comprised between −4.2 and −3.1[21].

Prior to the experiment, the average permeability of the fault zone and intact rock matrix were estimated from in situ hydraulic pulse tests at ~7 × 10⁻¹² and 1 × 10⁻¹⁷ m², respectively[20]. The initial state of stress was estimated at $\sigma_1 = 6 \pm 0.4$ MPa (sub-vertical), $\sigma_2 = 5 \pm 0.5$ MPa and $\sigma_3 = 3 \pm 1$ MPa (subhorizontal)[11]. Thus, the maximum injected pressures into these experiments represent an extreme pressurization level relative to the stress state, facilitating pronounced opening and slip. Based on laboratory velocity-step friction experiments on powdered (grain size < 125 µm) simulated gouge material collected from rock samples drilled from the fault zone, the frictional behavior evolves with increasing fluid pressure from rate-weakening at slow slip rate (<10 µm/s) to rate-strengthening at greater slip rate, and the initial friction coefficient was measured between 0.53 and 0.65[19].

**Experimental data**. Figure 2 present results for the three fluid injection experiments. Data exhibit similar evolution whatever the test although absolute values are different. In the early stage of each experiment, first, the fluid pressurization saturates the borehole interval isolated between the two inflatable packers, and elastically deforms the fault over the first tens of seconds. Then, each increasing pressure step causes clear fault opening and sudden slip acceleration followed by a slower creep. When slip accelerates suddenly because of a new pressurization step, the slip velocity increases typically from few µm/s up to 25 µm/s in these experiments (Supplementary Figs. 2 and 3). The final fault opening and slip reach, respectively, 41.3 and 41.8 µm (Test 1), 49.5 and 38.1 µm (Test 2), 65.7 and 110 µm (Test 3).

Data also show that fault slip is aseismic at injection, consistently with previous studies of the site[11,21] and with laboratory tests conducted on core samples drilled from the fault[19]. Indeed, seismic events are triggered after a certain amount of fluid pressurization and slip is accumulated over the fault, that is after 1391 s (Test 1), 177 s (Test 2), 679 s (Test 3) of injection into these experiments (Fig. 2B). For each experiment, seismicity is located at a distance of injection of 0.76–10.7 m (Figs. 1B and 2B). Duboeuf et al.[21] showed that the seismicity observed during these tests is located on fault segments and fractures in the damage zone surrounding the main pressurized fault planes.

The observation that the fault opens and slips during the injection implies that permeability and friction evolve during the fluid pressurization. Thus, we use the fault displacement data to quantify the frictional and permeability response. In the following, we first focus on the permeability evolution over the complete slip sequence, and second, on the transient changes in friction and permeability during an individual fast-increasing pressure step.

**Fault permeability evolution with cumulative slip at steady-state pressure conditions**. Considering the average permeability of the intact rock matrix is small (1 × 10⁻¹⁷ m²) compared to that of the fault (~7 × 10⁻¹² m²) and that injections are of relatively short duration, we assume that the fluid flow is mostly through the fault plane. During slip, the apparent fault permeability is governed by the evolution of fault aperture associated with dilation or compaction[23,24]. Consequently, the change in fault permeability relative to the initial fault permeability ($k/k_0$) can be related to the change in hydraulic aperture relative to the initial fault aperture ($\Delta b/b_0$) as proposed by Zhang et al.[25]:

$$\frac{k}{k_0} = \left(1 + \frac{\triangle b}{b_0}\right)^2 \qquad (1)$$

This equation assumes a Darcy's flow with an incompressible fluid of uniform density along a single smooth fault (i.e. no roughness). Permeability changes are estimated from the steady-state fluid pressure conditions, that is when the pressure is stabilized towards the end of the step-increasing phase and the fault displacement velocity is near zero (see Methods; Supplementary Fig. 4). These conditions allow an accurate estimation of permeability changes. We assume an average initial aperture ($b_0$) of 10 µm, consistently with previous permeability estimations[11,20].

Figure 3A, B show that fault permeability and fault-parallel displacement generally increase in response to step-increased fluid pressure, and the associated reduction in effective normal stress (i.e. total normal stress minus the fluid pressure). At the maximum pressure, and hence very low values of effective normal stress, the cumulative permeability increase is 26 (Test 1), 28 (Test 2) and 58 (Test 3) times the initial permeability. Permeability also increases with increasing slip (i.e. fault-parallel displacement in Fig. 3C). Interestingly, permeability evolution with cumulated slip exhibits a common behavior, which follows a same non-linear trend, independently of the test location in the fault zone, the orientation of the slipping plane, the initial stress state and the pressure stepping loading path. The permeability first increases of about 1.2 order of magnitude relative to the initial value over the first 20 µm of slip, and then a slope change in the permeability increase occurs with a smaller additional enhancement up to 1.58 order of magnitude (Test 3) over a maximum prolonged slip of 110 µm at the end of injection. Data also show that 60.8 (Test 1), 19 (Test 2) and 55.4% (Test 3) of the total permeability increase is achieved during the aseismic slip period before seismicity starts with the continuation of injection (Fig. 3C).

To interpret the whole non-linear evolution of permeability as a function of slip, we use an expression for the slip-dependent hydraulic aperture ($b_{slip}$) analogous to the dilatancy relationship of Van Den Ende et al.[26] defined from laboratory-derived microphysical model:

$$b_{slip} = \frac{b_{\max} - b_0}{1 - \gamma}\left[1 - \exp\left(-2\beta\frac{u_s}{L}(1 - \gamma)\right)\right] \qquad (2)$$

where $b_0$ is the initial hydraulic aperture before the fault is reactivated, $b_{max}$ is the maximum attainable aperture, $\beta$ is a constant that represents how much dilatancy is involved when the fault slips, $\gamma$ is a geometric constant, $u_s$ is the slip, and $L$ is a characteristic thickness of the fault. In the model, $u_s$ is sensitive to the change in effective normal stress, and so slip increases upon an increase in the fluid pressure as observed experimentally (Fig. 3B).

Using Eqs. 1 and 2, we conduct an iterative grid search approach to find the values of the parameters $b_0$, $b_{max}$, $\beta$, $\gamma$ and $L$ that minimize the misfit between model predictions and observed permeability evolution as a function of slip. The inversion was performed for different sets of initial values for the five parameters. The bounds on the prior distribution are given by $5 < b_0 < 15$ µm, $60 < b_{max} < 80$ µm, $0.1 < \beta < 5$, $5 \times 10^{-5} < \gamma < 10 \times 10^{-5}$, and $60 < L < 250$ µm. The misfit

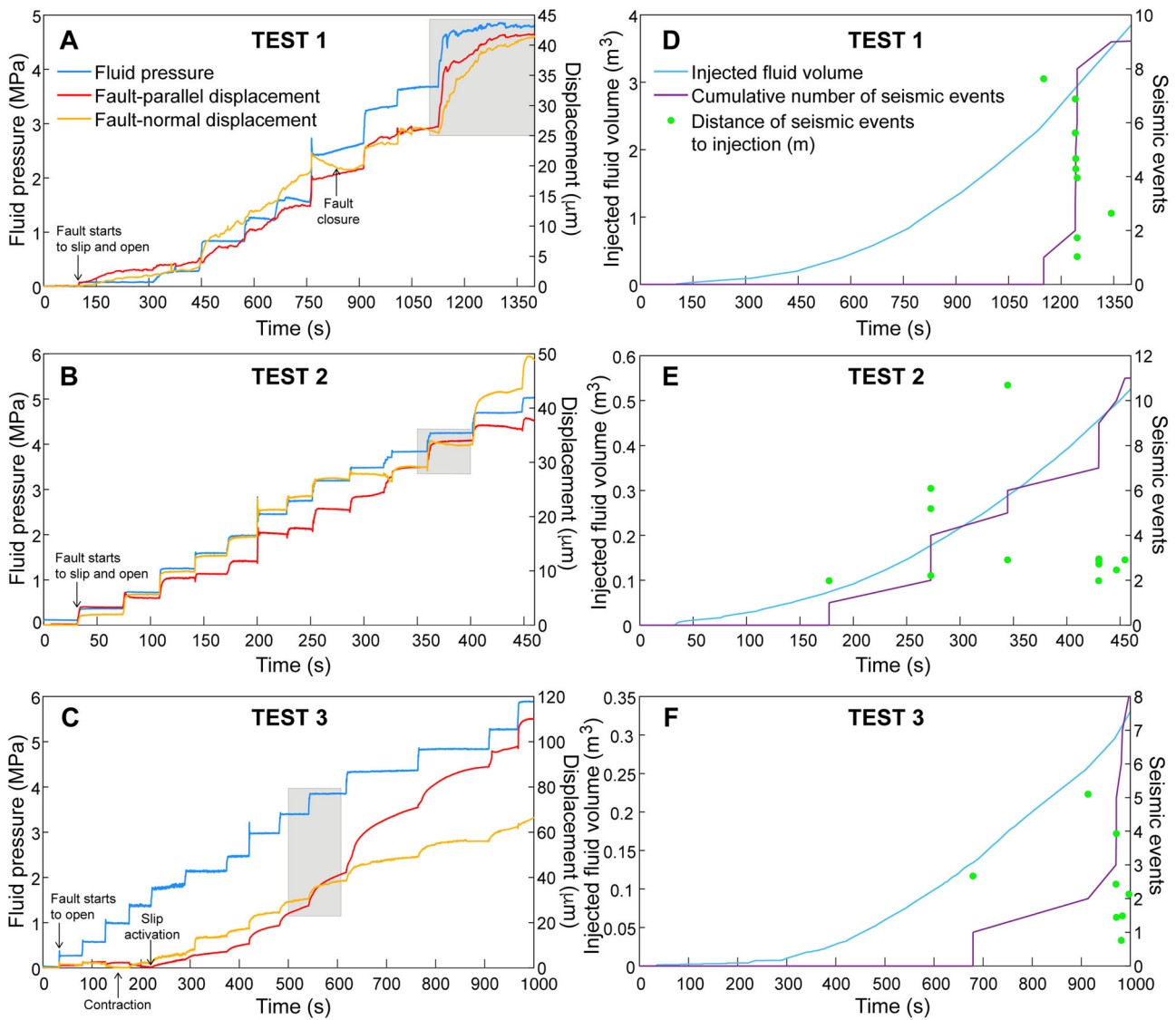

**Fig. 2 Experimental data collected during three fluid injection tests (Tests 1, 2 and 3) within the fault zone. A–C** Fluid pressure (blue), fault-normal (opening in orange) and fault-parallel (slip in red) displacements; (**D–F**) Injected fluid volume (cyan line and left axis), cumulative number of seismic events (purple line and right axis), and distance (m) of seismic events relative to injection (green dots and right axis). The gray boxes in the panels (**A**, **B**, and **C**) represent the data sequences used for the modeling.

between the observations and predictions is estimated with the reduced chi-square ($\chi_r^2$). We find a best-fit solution (Fig. 3C; $\chi_r^2 = 0.3$) with values of $b_0 = 7.35\,\mu m$, $b_{max} = 62.9\,\mu m$, $\beta = 2.1$, $\gamma = 8.12 \times 10^{-5}$ and $L = 191.25\,\mu m$. The model solution fits well the whole dataset with reasonable values. This shows that the permeability variation related to the complex in situ fault hydromechanical response to reducing effective normal stress can be reasonably figured by relating the hydraulic aperture to dilatant slip as observed at the laboratory scale[27,28].

**Transient frictional and permeabiity response during a fast-increasing pressure step**. As seismicity and aseismic slip are controlled by the fault frictional and hydraulic response, we use the measured fault slip and opening to explore the transient evolution of friction and permeability during a step-increased pressure. Data show that changes in fluid pressure result in an immediate acceleration of slip followed by a gradual and slower phase of slip (Fig. 2). Such evolution of fault slip has been observed in laboratory experiments studying the effects of sudden changes in normal stress on slip and frictional strength using bare

rock surfaces and layers of fault gouge in dry conditions[29,30] or during fluid pressurization[27,31]. Upon a step change in normal stress, the transient slip response was traditionally reproduced with the rate-and-state friction law[4,32]. Here, we examine whether our in situ data collected in decametric scale experiments can fit this type of friction law during a stepped-pressure increase. We evaluate the data using a 1D spring-slider model with a uniform fluid pressure to retrieve frictional parameters. This assumption is valid for high initial permeability fault[26], such that the characteristic timescale of fluid diffusion is smaller than the timescale of deformation. We selected three independent sequences in Tests 1, 2, 3 (Fig. 2, and Supplementary Figs. 2, 3 and 5). These sequences selected at the higher-pressure steps have similar initial pressures of 3.67 (Test 1), 3.83 (Test 2) and 3.4 MPa (Test 3) before the step-increase, and a slip that is well established on the fault. Additionally, these sequences were associated to seismicity in Tests 1 and 2 (Fig. 2A, E), and an aseismic period for Test 3, allowing a comparison between different modes of fault slip. We used the measured fluid pressure data as loading path in the model. In order to have the clearest possible stepped-pressure

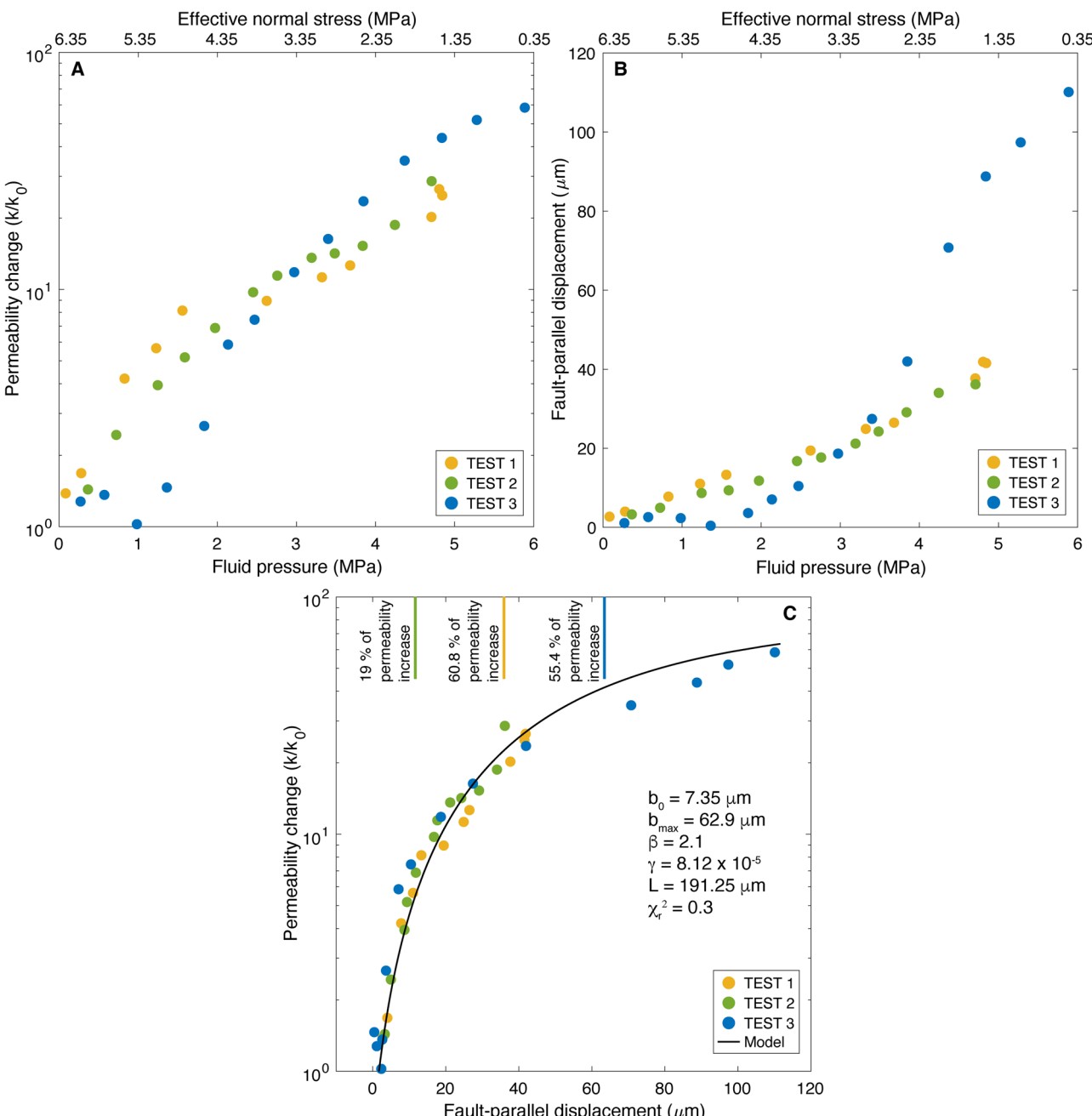

**Fig. 3 Evolution of fault permeability during fluid pressurization. A** Permeability change ($k/k_O$) and (**B**) fault-parallel displacement (i.e. "cumulated slip") as a function of effective normal stress and increasing fluid pressure. **C** Permeability change ($k/k_O$) versus cumulated fault-parallel displacement. The black line in (**C**) is the best-fit modeling result obtained with Eqs. 1 and 2. The reference values for the parameters $b_O$, $b_{max}$, $\beta$, $\gamma$, $L$ are indicated and allow to fit in a single model the whole permeability evolution measured in the three tests. The vertical lines indicate the onset of seismicity and the percentage of permeability increase achieved during the aseismic slip period, respectively observed during Test 1 (orange), Test 2 (green), and Test 3 (blue).

perturbation, we have intentionally chosen short sequences (tens of seconds) during the course of injection when the fault motions are well developed, and the magnitude of slip and normal opening is comparable.

The velocity dependence of friction ($\mu$) was interpreted with the rate-and-state friction law[4,32]:

$$\mu = \mu_0 + a\,ln\left(\frac{V}{V_0}\right) + b\,ln\left(\frac{V_0\theta}{d_c}\right) \tag{3}$$

where $\mu_o$ is the friction coefficient at a reference slip velocity ($V_0$). The parameter $a$ quantifies the direct effect of a change in slip velocity ($V$). The parameter $b$ describes the effect of the state

variable $\theta$. The characteristic slip distance, $d_c$, governs the evolution of the state variable. We use the Dieterich law (also, called "Aging" law[33]) extended by Linker and Dieterich[29] to account for variable effective normal stress in the formulation of the state variable as:

$$\frac{d\theta}{dt} = 1 - \frac{V\theta}{d_c} - \alpha\frac{\theta}{b(\sigma_n - p)}\frac{d(\sigma_n - p)}{dt} \tag{4}$$

where $\sigma_n$ is the normal stress, $p$ is the fluid pressure, and $\alpha$ describes the evolution of shear stress associated with changing effective normal stress ($\sigma_n$-$p$). The choice of this evolution law is

motivated by previous studies showing a good fit to experimental data[34].

To take into account the medium stiffness ($K_s$) and its elastic coupling with the fault, Eqs. (3) and (4) are coupled with a single degree of freedom relationship assuming homogeneous shear stress and slip distribution on the fault as:

$$\frac{d\tau}{dt} = K_s \left( V_{lp} - V \right) \qquad (5)$$

where $V_{lp}$ is the load point velocity.

We modeled the fault slip data by solving Eqs. (3), (4), and (5) simultaneously using a fifth-order Runge-Kutta method. Parameters $a$, $b$, $d_c$, $\mu_0$, $\alpha$ and $K_s$ were obtained by an iterative grid search inversion approach (Supplementary Table 1). As the fault is inactive prior to injection, the model is not loaded tectonically (i.e., $V_{lp} = 0$). We impose the initial normal and shear stresses previously estimated in the field[11,21]. 

Furthermore, using the calculated slip and slip velocity from the previous spring-slider model, we then assess the transient change in fault aperture. Fang et al.[5] and Ishibashi et al.[6] showed from the analysis of laboratory experiments of stepped-velocity friction tests with fluid infiltration in fractures that the evolution of aperture is dependent on both the effects of slip and slip velocity. Friction and permeability of faults are known to change during slip due to the rearrangement and destruction of asperities. Permeability may increase due to shear induced dilation[6] or decrease as a result of gouge formation[35] or irreversible changes in surface roughness[36]. During slip, frictional strength is affected by the state of the contact area between the displacing surfaces[4,32,33]. Previous field experiments on faults have also demonstrated that the contribution of slip on the evolution of permeability and friction is central[11,13,16,37]. This gives a degree of confidence about the applicability of slip- and slip velocity-dependent models inferred from laboratory data for the analysis of in situ data. Analogous to the shear dilation relationship of Samuelson et al.[38], Fang et al.[5] proposed the following two equations to represent the evolution of aperture ($b_{evo}$) due to a dilation parameter dependent on the slip velocity ($\Delta v$) and slip-dependent aperture ($b_{slip}$):

$$\Delta v = \psi \cdot ln\left( \frac{V^{i-1}}{V^i} \left[ 1 + \left( \frac{V^i}{V^{i-1}} - 1 \right) \cdot e^{-V^i \cdot t^i / d_c} \right] \right) \qquad (6)$$

$$b_{evo} = b_{slip}(1 + \Delta v) \qquad (7)$$

where the index $i$ refers to the $i$th velocity step, $\psi$ is a dilation factor, and $t$ is the time. We use the best-fit solution to fault slip, together with slip velocity, as input of the aperture model to estimate the resulting permeability using Eqs. (1), (2), (6) and (7). In Eq. 2, $b_0$ is taken before the pressure step, while $\beta$ is adjusted around the value previously deduced at steady-state conditions (Supplementary Table 2). $b_{max}$, $\gamma$ and $L$ have the values inferred in the previous analysis (see section 4 and Supplementary Table 2).

Figure 4 shows the numerical solutions that fit reasonably well the experimental data, and the associated friction and permeability response during the pressure step. Interestingly, most results are plausibly described with rate-and-state friction coupled with permeability change, but some discrepancies also show that this is not systematic for all cases.

For Test 1 (Fig. 4A, B), overall, the slip is relatively well reproduced, notably the sudden slip at the initiation of pressure step, which causes a rapid friction increase ($\Delta\mu$) of 0.054 followed by a slow decay to a new steady state value, $\mu_{ss} = 0.478$. For the secondary phase of slower slip, the model reproduces the general evolution; however, some differences appear between model and data. Indeed, the measured slip oscillations could reflect the

interplay between different fault segments within the damage zone, an effect that is not considered in our single fault model. This best-fit solution ($\chi_r^2 = 0.594$) is obtained with rate-weakening frictional properties (Supplementary Table 1). At the same time, a discrepancy between measured and calculated permeability (Fig. 4B) is observed, resulting in a permeability evolution ($k/k_0 = 2.5$) that is poorly coupled with slip and slip velocity at the considered time scale. The exact process responsible for such a difference remains subtle. It could be related to additional hydromechanical effects or geometrical complexities ignored in the model we use. In such well-developed fault zone, it could also reflect heterogeneous fluid flow (i.e. channeling) over the pressurized fault segment or a fluid leakage from the pressurized fault into the surrounding connected fractures or bedding planes. As the injected fluid volume in Test 1 is much larger than for the other tests, it is likely that an extended network of faults is pressurized, leading to more complex hydromechanical response.

For Test 2, the fit to the slip is excellent ($\chi_r^2 = 0.018$) with rate-weakening frictional parameters exhibiting a direct friction increase ($\Delta\mu = 0.057$) followed by a slower decay to $\mu_{ss} = 0.422$ (Fig. 4C, D, Supplementary Table 1). Moreover, the first increasing portion of permeability change is well reproduced at the initiation of pressure step, whereas the second declining portion of slow reduction is not well matched by the model. To produce a reasonable fit to the data, a value of $\beta = 2.8$ and $\psi = 2.2$ is required. This results in a maximum permeability change ($k/k_0$) of 1.36.

For Test 3 (Fig. 4E, F), the numerical solutions fit well both the slip ($\chi_r^2 = 0.061$) and permeability data. Friction obeys to a rate-strengthening behavior with a sudden friction increase ($\Delta\mu = 0.048$) immediately at the pressure step and then follows by a slower decay to $\mu_{ss} = 0.523$. The fit to the transient change in permeability is excellent (Fig. 4F) and results in a maximum permeability change ($k/k_0$) of 1.76.

In summary, models tend to show two sequences for the permeability change (Fig. 4B, D, F). First, at the initiation of pressure step, a sudden permeability increase is observed in association with an accelerating slip. Then, when the pressure tends to stabilize, fault slip decelerates and permeability enhancement or reduction is slower and more gradual. Interestingly, the permeability and friction evolve concurrently under conditions of varying stress. This observation suggests a direct link between the transient change in permeability and friction presumably through rearrangement of fault surface asperities as a result of slip and opening or closure. The comparison between experiments shows two fault behaviors. In the first case, a rate-weakening behavior is associated with a fault closure after the pressure increase (Test 2), and in the second case, a rate-strengthening behavior is observed together with a fault opening (Test 3).

Although some differences in the fault response observed between the tests can be due to the different levels of fracturing of the rock matrix that surrounds the pressurized fault plane, the exact mechanical process responsible for permeability increase or decrease after the sudden acceleration remains elusive. Based on geological observations of the borehole intervals before injection, they could reflect geometrical changes of contact areas, gouge content and void space that comprise the fault during deformation at low effective stress for Tests 1 and 2, whereas the hydromechanical response observed in Test 3 may be related to the overall permeability of the fractured damage zone surrounding the tested fault segment.

Overall, the best-fit model parameters (Supplementary Table 1) indicate a range of frictional values, $\mu_0 = 0.5$, $a-b = -0.028$ to $0.01$, $d_c = 5$ to $50\,\mu m$, and $\alpha = 0.091$ to $0.1$, that are reasonably

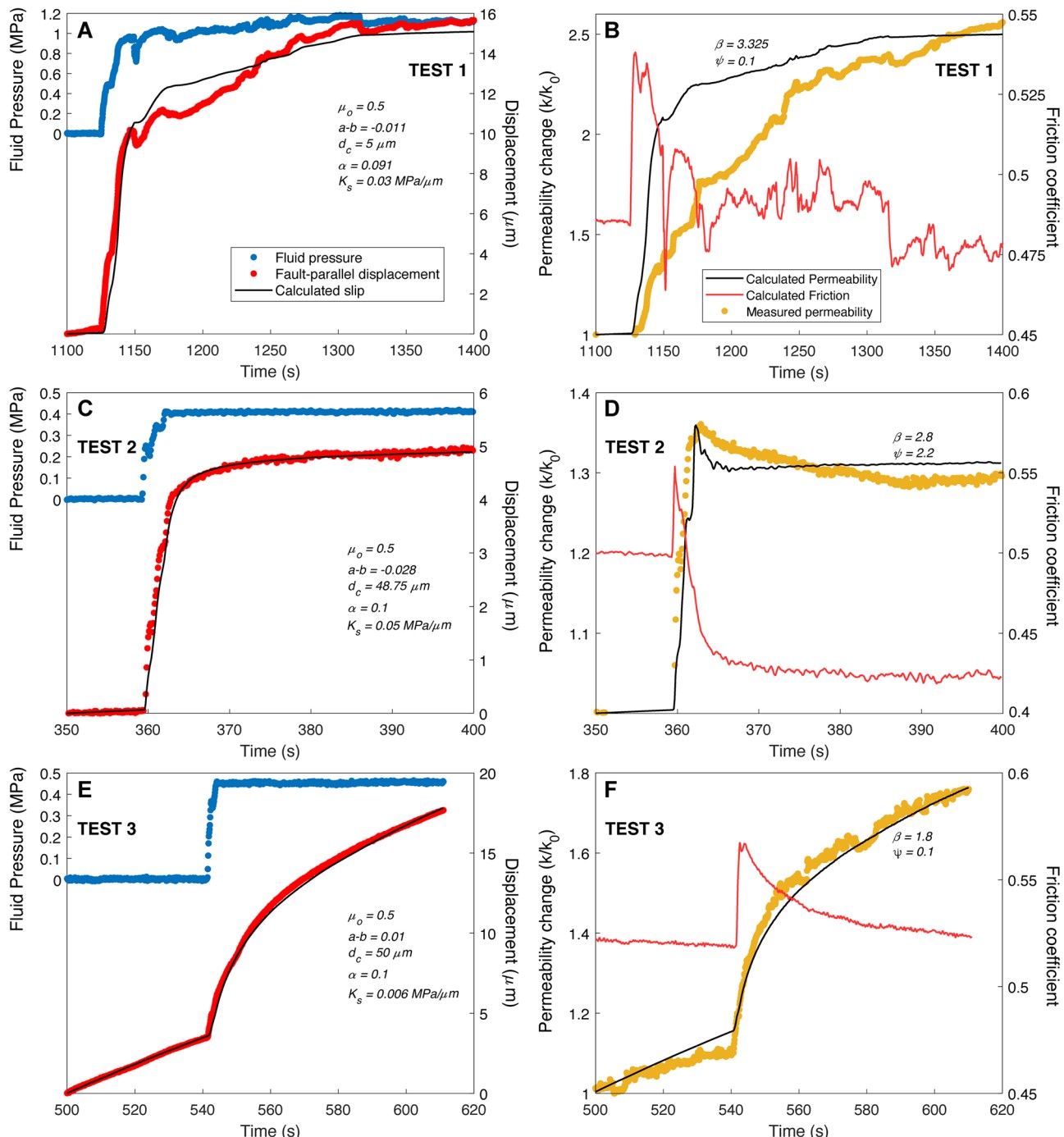

**Fig. 4 Transient evolution of fault slip, permeability and friction during a fast-step increase in fluid pressure.** Best-fit model solutions to experimental data and associated change in friction and permeability for (**A–B**) Test 1, (**C–D**) Test 2, (**E–F**) Test 3. Frictional parameters $a$, $b$, $d_c$, $\mu_0$, $K_s$ and $\alpha$ were estimated using Eqs. (3), (4), (5), and parameters $\beta$, $\gamma$, and $\psi$ for permeability change using Eqs. (1), (2), (6) and (7).

consistent with previous laboratory measurements of rate-and-state friction parameters ($\mu_0 = 0.55$ to $0.65$, $a−b = −0.01$ to $0.0056$, $d_c = 1.2$ to $59.6\,\mu m$) on centimeter-sized gouge samples collected in the same fault zone[19].

## Discussion

In this work, we studied the evolution of fault permeability and friction in a natural fault activated by fluid injection. We performed three experiments with direct observations of the fault hydromechanical response at the decameter scale in a low stress regime. It is important to note that our experiments represent a

case where the fault zone has initially a high permeability, and, during injection, it corresponds to a slowly slipping fault.

Through our investigations, the data show that the increase of fluid pressure strongly influences the transient evolution of fault permeability and friction parameters, and consequently, the fault stability. First, we find that the fault permeability estimated at steady-state pressure conditions enhances significantly with reducing effective normal stress and injection-induced cumulated slip. The permeability evolution with cumulated slip is non-linear, consistently with previous experimental studies at the laboratory scale of fluid flow in fractures under shear load[39]. Interestingly,

the relation between permeability change and slip shares a common macroscopic behavior regardless of the test location in the fault zone, the orientation of the slipping plane, the initial stress state and the pressure history. In such initially high-permeability fault, the fluid pressurization is a driver of dilatant slip, in addition to poro-elastic opening that appears limited in these experiments. Furthermore, our data shows that most of the fault permeability increase occurs during the period of aseismic slip into the three experiments, similarly to other studies[11,37], which showed that the seismicity is localized far from the injection point where the hydromechanical measurements are performed.

Moreover, our estimates of the critical nucleation lengths for seismic rupture to start (see Methods) are comprised between 0.54 to 9.74 m for Test 1, and 2.06 to 37.3 m for Test 2, which prevent seismicity to occur near injection. Additionally, the small values of critical slip distance estimated at the injection points favor rapid strength drop at this place. As a result, stress concentration in the surrounding is large enough to trigger seismicity away from the pressurized zone. Thus, the seismicity observed in these experiments represents a remote response of the fractured rock volume to the forcing caused by aseismic deformation of the pressurized plane.

At the short time scale of a fast-pressure increase, we also observed that in some cases the transient change in permeability is well coupled to slip, and, in other cases, it cannot be explained only by slip and/or slip rate. Nevertheless, although we obtain a reasonable fit to data with a simple model, other coupled processes, not considered here, can also influence the fault response, such as the deformation associated (1) with geometrical irregularities heterogeneously distributed along natural faults, (2) with the fractures network in the volume surrounding the pressurized fault segment, or (3) with the fluid pressure gradient developing from the injection point toward the outer fault boundaries. Indeed, owing to the interplay between permeability and friction, the fault hydromechanical response may result in heterogeneous fluid pressure distribution, channeling flow, stress interaction between neighboring asperities or fractures, and a variety of slip behavior. Moreover, in some conditions, the permeability enhancement is not well coupled with aperture changes and can rather be linked to flux-driven particle mobilization that induces clogging and unclogging within the fault[24]. Thus, for Tests 1 and 2, the difference between observed and calculated permeability enhancement may be explained by a combined results of both slip-induced dilation and fluid flow-driven particle mobilization.

Although, fault slip is a complex process dependent on different factors, including fault type, host rock lithology, physical properties and stress conditions, our observations and models demonstrate that the transient evolution of fault friction and permeability is both linked to effective normal stress and slip. This co-evolution can be explained consistently in the framework of laboratory-derived rate-and-state friction law together with a permeability model considering the effects of both slip and slip velocity[5,6,23,40,41]. Overall, the frictional parameters derived from the in situ experiments are in good agreement with those measured in the laboratory, despite the scale difference at which the parameters were estimated, from centimeters in the laboratory to meter in the field. Thus, the measurements of fault displacement during in situ fluid injection experiments allow to validate with direct observations the physical laws and processes derived at the laboratory scale, and hence, allow for their extrapolation to geological reservoirs and seismogenic zones.

## Methods
**Estimate of the fault permeability change from the temporal evolution of fault-normal displacement**. To estimate the permeability change of the fault from

the temporal evolution of fault-normal displacement measured at the injection, we assume that the change in hydraulic aperture is equivalent to the change in normal displacement, consistently with previous experimental works of the hydro-mechanical response of permeable fractures in in situ conditions[42,43]. Thus, we used our direct measurements of the fault-normal displacement (i.e., "opening"), and selected a stabilized value at near steady-state fluid pressure conditions, that is when the fluid pressure stops varying towards the end of the transient step increase, and before the next pressure step (Supplementary Fig. 4). This choice allows a sufficient stabilization time for near steady-state conditions to establish at the injection, with limited storage and poro-elastic effects in the injection chamber, and absence of chemical effects which generally occur over hours to days, a longer time compared to the short duration (from 450 s to a maximum of 1400 s) of the injection tests studied here. To summarize, the permeability change (Fig. 3 and orange line in Fig. 4) estimated from experimental data is directly calculated from the measured opening caused by increasing fluid pressure using Eq. 1.

**Estimate of the nucleation length for seismic rupture to start**. To estimate the nucleation length for seismic rupture to start along a rate-and-state fault, the nucleation theory indicates that an earthquake occurs within rate-weakening regime ($a-b > 0$) once the slipping region reaches a critical size ($L_c$)[32]:

$$L_c = \frac{G \, d_c}{(\sigma_n - p)(b - a)}$$

where $G$ is the rock shear modulus (here, G = 7.5 GPa). For the estimated rate-weakening parameters at the injection point, $d_c$ ranges from 5 μm (Test 1) to 48.75 μm (Test 2), $(b-a)$ ranges from 0.011 (Test 1) to 0.028 (Test 2), and $(\sigma_n$-$p)$ ranges from 0.35 MPa for the maximum pressures to about 6.35 MPa at the initial zero pressure. For this range of values, our estimates of the critical nucleation lengths are comprised between 0.54 to 9.74 m for Test 1, and 2.06 to 37.3 m for Test 2.

Over the fault, slipping zones with radii ($L$) greater than the critical nucleation length $L > L_c$ are susceptible to trigger seismicity whereas those with radii $L < L_c$ are not. This analysis is consistent with the slip being aseismic in the pressurized area as observed in situ with seismic events located between 0.76 and 10.7 m in the volume around the injection point. Moreover, the source radius of the seismic events (0.19 to 0.57 m) estimated in[21] are smaller than the theoretical minimum length over which earthquake nucleation can occur.

It is also important to note that the frictional and elastic properties in the zones of seismicity observed at a distance of injection points are not evaluated in the present study. A future characterization of the frictional parameters within the whole fault zone may help for better defining the seismic potential of fault segments and fractures around the injection.

## Data availability
The data supporting the analysis and conclusions are available in supporting information.

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

## Acknowledgements

This study was supported by the French government through the ANR HYDROSEIS project funded by the National Research Agency (ANR) with the reference number ANR-13-JS06-0004-01.

## Author contributions

F.C. designed this study and carried out the modeling. All the authors contributed to the experimental design and performed the experiments. All authors analyzed the results and wrote the paper.

## Competing interests

The authors declare no competing interests.

## Additional information

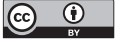

