## [Peer Review File · Nature Communications]

**REVIEWER COMMENTS**

**Reviewer #1 (Remarks to the Author):**

**This paper uses borehole fluid pressurization tests directly into a fault zone in an underground**
**laboratory, in order to measure permeability (via the fluid pressure), dilatancy (aperture opening), and**
**slip on the fault in response to fluid pressure steps. If I understand correctly, in addition to the direct**
**measurements, permeability and slip functions are also calculated from the dilatancy measurements.**
**Then, expressions are combined to relate the permeability to the slip. At the same time, slip**
**measurements are combined with the rate-and-state friction law to produce a friction record during the**
**pressure step. The outcome is, for a given pressure step, a calculated permeability function that can be**
**compared to the measured permeability, a calculated slip function that can be compared to the**
**measured slip, and a calculated friction function.**

**In general, I think the approach is sound and the work is clearly important for the topic of induced**
**seismicity. The measurements seem well done and I don't see anything I would consider to be a**
**serious methodological error. I did find the manuscript confusing at points, and I think the biggest area**
**of improvement would be more precise explanations and language, which I list below. Some points**
**may seem picky but I think clarity is important, because it may affect the interpretations. I don't think**
**this will affect the story much, so the paper should be publishable after revision.**

**I think the role of effective normal stress should be emphasized more. Increasing the fluid pressure of**
**course reduces the effective normal stress, here to very low values. This is really the cause of all**
**subsequent observations, for example permeability, which is known to be sensitive to effective normal**
**stress. I would like to see this in the text, and the effective normal stress can easily be added to Figure**
**3 as a double x-axis. Also, it is the effective normal stress that appears in the rate-and-state friction**
**formulation. It is just called "normal stress" in the Hong and Marone and Linker and Dieterich papers**
**because those experiments were conducted room dry with no pore fluid (so that the normal stress is**
**the effective normal stress).**

**I think the fault zone itself needs to be more precisely described. It is introduced as "the fault zone"**
**but is actually made of multiple fault strands (principal slip zones) and a damage zone, and within the**
**damage zone there are "faults and fractures", so it is difficult sometimes to keep this all straight. What**
**do the three boreholes actually intersect? It looks like Tests 1 and 2 are performed directly on one of**
**the principal slip zones, whereas Test 3 is in the damage zone. Are both Tests 1 and 2 performed on a**
**clay-bearing fault? This is of course important because clays have very different hydromechanical**
**properties compared to fractured limestone. Is there a significant difference between the location of**
**Tests 1 and 2? If so, this could explain some of the differences seen in Figure 4, for example the**
**permeability fit in Test 3 may be related to the large overall permeability of the damage zone, whereas**
**the less well-fit permeability model in Tests 1 and 2 could be the effect of clay gouge. But if the**
**location of Tests 1 and 2 are the same, that does not explain the difference between these two.**

**I'd like a bit more information on the measured permeability values. What exactly is meant by the**
**permeability of the "rock matrix"? Intact limestone, or the damage zone which includes fragments of**
**intact limestone and a network of fractures? What is meant by the permeability of the fault zone?**
**Across-fault permeability? Along-fault permeability? Is this a fault with clay gouge?**

**I found the calculations and relation of permeability to slip to be somewhat convoluted. Both the**
**permeability increase and slip activation are the result of dilatancy (which is of course the result of**
**reduced effective normal stress). So I am not sure if it is really correct to say that the permeability is a**
**function of the slip (i.e., they are both effects of the same process but one does not necessarily cause**
**the other) (e.g. Line 18). For the calculated permeability shown in Fig 4, why calculate it from the slip,**
**which is calculated from the aperture opening, and not directly from the aperture opening itself? I**
**think it also would help if the main goal/result of the study is clearly stated, "co-evolution of friction**
**and permeability is important" is not very clear. I guess the goal is to be able to calculate permeability**
**and friction and/or frictional stability from a known pressure perturbation and measured dilatancy**
**and/or slip? I suggest stating something like this directly.**

**Care should be taken when referring to "seismic slip" vs "aseismic slip". All the seismicity is off-fault**
**(or at least remote from the test location in the borehole. So really, all the measured slip here is**
**actually aseismic, and the "seismic slip" is triggered on a different fault somewhere else. On a related**
**note, on Lines 288-290 it is stated that "small values of slip-weakening distance" (I guess here the**

critical slip distance D_c is meant?) are suggested to be responsible for remotely triggered seismicity,
but the RSF parameters and critical nucleation lengths obtained here are only relevant for the fault for
which the slip was modeled. For remote seismicity, the RSF and elastic properties for that fault have to
be evaluated.

Care also needs to be taken with the term "evolution of friction", I guess you mean here the friction
time series as shown in Fig 4. But the friction parameters that determine fault stability (the rate-and-
state friction parameters) are fixed for a particular modeled pressure step, and do not evolve, or co-
evolve with permeability (e.g. Line 22).

**Line 87:** what samples were used in these experiments? Limestone? Clay gouge?

**Line 107:** so the seismicity occurs only in the damage zone, and not on any of the clay-lined principal
slip zones? this makes sense based on what is known about clay friction in general but is worth
mentioning

**Lines 218-223:** $\Delta\phi$ is described as a "dilation parameter dependent on slip velocity" but ϕ is
conventionally porosity, so is $\Delta\phi$ just a porosity change? It seems simpler to describe it this way.

**Lines 264-267:** it would be helpful to report the rate-and-state friction values from the previous study,
so the reader can directly compare.

**Line 311:** as noted above, they are both linked to dilatancy (aperture opening) and reduced effective
normal stress

**Reviewer #2 (Remarks to the Author):**

In their manuscript "Co-evolution of permeability and friction in a slowly slipping fault activated by
fluid pressurization", Cappa et al. presented the results of three in-situ injection experiments where
fluid pressure, fault displacement (normal and parallel) and slip velocity are measured. They then used
a numerical model relating the hydraulic aperture (assumed to be equal to fault-normal displacement),
permeability, slip, slip velocity, and rate-and-state friction to fit the observed permeability and fault
slip data, finding a reasonably good fit. This study corroborates with some previous experimental
studies they and others have done, showing the importance of permeability enhancement upon slip
triggering. The results are very interesting to the seismology community, especially for those modeling
injection-induced aseismic slip. I do have a few concerns regarding the data fit, and would appreciate if
the authors could address the following comments.

**Main comments**

**1. The fit to Test 1 vs. Tests 2 & 3 show some rather significant differences. The reason why Test 1**
**does not have as great fit is explained in the Discussion, but when it was first mentioned in Lines 242-**
**246, the explanation was rather vague and general, perhaps adding some explanations there will make**
**the reader more convinced, and then in the discussion, those points can be further developed.**
**However, in concept, all tests involve similar set-ups, so whatever has not been taken into account in**
**the model for one test also wasn't taken into account for the other tests. So why is there much less**
**discrepancy for the other two tests?**
**2. I'm particularly interested in the iterative parameter search to produce the best fit. The values of d_c**
**and K_s , for example, span over an order of magnitude across the different tests, which are very close to**
**each other in space. Is it realistic to have them be so different for such similar test locations?**
**3. Is there any consideration of using the mass balance equation to fit the fluid pressure evolution**
**using the parameters found above? Maybe that will be a good check on whether they are not just**
**(over)fitting the permeability profile, if they can also reasonably fit the pressure profile.**
**4. My concern with the hydraulic-aperture and permeability relation is that, if the fault has opened so**
**much to produce the observed order of magnitude enhancement in permeability, then why is there no**
**obvious dilatant suction effect seen in the pressure profile?**

**Minor comments**

**1. Equation (1) the cubic law is written as square**

- **2. Line 152: do the constants beta and gamma have meanings?**
**3. Line 178: could you explain why this assumption is valid for high initial permeability fault?**
**4. Lines 182-183: do Tests 1 & 2 show induced seismicity? The slip rates as shown in supplementary**
**Figure 2 are quite low for all cases and never reach seismic speeds?**
**5. Lines 262-263: could you expand on what is interesting about the permeability and friction evolving**
**concurrently?**
**6. Figure 2: the green dots are labeled as distance of seismic events to injection, but where is the scale**
**that shows how many far it is to injection?**

**Reviewer #3 (Remarks to the Author):**

**A review of "Co-Evolution of Permeability and Friction in a Slowly Slipping Fault activated by Fluid**
**Pressurization by Cappa F., Giglielmi Y, De Barros L.**

**This manuscript presents a large-scale in-situ experiment on permeability and friction evolution in a**
**natural fault. The authors performed fluid injections in a km-scale fault zone in limestone in the LSBB**
**lab (France). They measured fluid pressure, normal and shear displacements, and seismic activity.**
**Permeability and friction are modelled from these data. This topic is highly relevant for unconventional**
**reservoir geomechanics. The main result presented by the authors is that friction and permeability are**
**dependent on both slip and slip velocity.**

**The paper is well written and the main messages contained in it are clearly delivered and well**
**supported by experimental and modelling data. It definitely deserves to be published. However, the**
**work would better fit in a more specialized journal, where part of the interesting supp. mat. could be**
**incorporated into the main text. I have only moderate comments.**

**Comments:**

**L14: in-situ (ita)**

**L31: Specify the frictions laws and permeability laws derived from lab tests**

**L41: Cite Zappone et al., 2021, solid earth?**

**L58-60, Fig 1: Add fault orientation, and movement on fig 1**

**L 66-72: Please give more details on injections protocols in supp. mat. For example, justify the**
**injection depth, durations, volumes, max pressures ect...**

**L 72: Similarly, give the locations of the seismic sensors (accelero, and geophones) in fig 1. r supp.**
**mat.**

**L90: Careful here! The lab experiments were performed on gouge material. The natural fault is rough at**
**different scales and without gouge material (L 62-63). It is well known that the dependences of friction**
**to slip, and slip rate are strongly related to fault structure (with (a-b) increasing with slip-velocity in**
**gouge material, and decreasing on bare surfaces, Noel et al., (2019)). Moreover, the static friction**
**coefficient of gouge material is different than the one for bare surfaces. Note also that gouge material**
**has almost no cohesion.**

**Fig 2: Add a marker at the beginning of the different deformation stages in Fig. 2: elastic deformation**
**stage, fault opening, slip acceleration, creep**

**L113: Is it the permeability or the transmissivity that is measured? How is the initial fault effective**
**thickness measured?**

**L120: How is normal fault aperture linked to hydraulic aperture?**

**L 125: Give the hypothesis behind equation 1: A single fracture is reactivated (i.e. interplay with**
**different fault segments is not taken into account), incompressible fluid of uniform density, steady**
**state pressure ect... Moreover, please acknowledge that this equation does not consider the complexity**
**of the aperture field due to contact roughness.**

**L132: It is not clear to me how fault parallel displacement is calculated. Is it calculated at the end of**
**each step?**

**L149: Related to previous comment, what are the hypothesis behind equation 2? If I am not wrong the**
**equation is derived from the CNS model developed for granular material. How does complex fault**
**roughness, geometry, multiple segments influence the link between dilatancy and slip? (with probable**
**complex fluid velocity field, slip, slip rate in all directions). Here, slip is measured only at the injection**
**point.**

**L173: Cite Noel et al., 2019, JGR?**

**L175: in-situ**

**L176: How have the 3 sequences been selected?**

**L217: Cite Samuelson et al., 2009?**

**L217: Remind the reader why friction and, even more importantly, permeability depend on slip and slip**

velocity in Fang's experiments (shear enhanced dilatation, change in contact distribution? ect...), and justify why this is also true (or not) in the in-situ experiments described here. Note that Acosta et al., have shown different results, such that shear reduces dilatancy. Dilatancy strongly depends on initial roughness and slip.

L 217: Is it possible that mechanisms such as flux-driven unclogging of the fracture possibly link frictional evolution to permeability change in the presented experiment?

L232/figure 4: I would be curious to see more of those pressure steps fitted by the model.

L262: Please explain why some fault portions do experience dilation why others see compaction.

L285: Remind the reader that seismicity is localized far from the injection point (where measurements are performed).

L287: Give L_c for test 3.

L330: Provide a zoom of several steps in supp mat. I would like to see the steady-state fluid pressure conditions, that seems not completely achieved in fig 4 of supp mat.

REVIEWER COMMENTS

- **Reviewer #1 (Remarks to the Author):**

This paper uses borehole fluid pressurization tests directly into a fault zone in an underground laboratory, in order to measure permeability (via the fluid pressure), dilatancy (aperture opening), and slip on the fault in response to fluid pressure steps. If I understand correctly, in addition to the direct measurements, permeability and slip functions are also calculated from the dilatancy measurements. Then, expressions are combined to relate the permeability to the slip. At the same time, slip measurements are combined with the rate-and-state friction law to produce a friction record during the pressure step. The outcome is, for a given pressure step, a calculated permeability function that can be compared to the measured permeability, a calculated slip function that can be compared to the measured slip, and a calculated friction function.

In general, I think the approach is sound and the work is clearly important for the topic of induced seismicity. The measurements seem well done and I don't see anything I would consider to be a serious methodological error. I did find the manuscript confusing at points, and I think the biggest area of improvement would be more precise explanations and language, which I list below. Some points may seem picky but I think clarity is important, because it may affect the interpretations. I don't think this will affect the story much, so the paper should be publishable after revision.

We are grateful to the reviewer for the positive comments on our manuscript, and the very interesting suggestions of improvement. We have tried to address all your questions and the details are below.

I think the role of effective normal stress should be emphasized more. Increasing the fluid pressure of course reduces the effective normal stress, here to very low values. This is really the cause of all subsequent observations, for example permeability, which is known to be sensitive to effective normal stress. I would like to see this in the text, and the effective normal stress can easily be added to Figure 3 as a double x-axis. Also, it is the effective normal stress that appears in the rate-and-state friction formulation. It is just called "normal stress" in the Hong and Marone and Linker and Dieterich papers because those experiments were conducted room dry with no pore fluid (so that the normal stress is the effective normal stress).

We agree with this point, and have added a new axis at the top of figures 3a and 3b to show the evolution of permeability and fault-parallel displacement as a function of the effective normal stress (i.e. total normal stress minus the fluid pressure). In the text, we have also added this point in lines 18, 150-152, 173, 184, 329 and 366.

This is correct the normal stress is the effective normal stress in the rate-and-state formulation used here to analyze our fault activation experiment cause by increasing fluid pressure (line 218).

I think the fault zone itself needs to be more precisely described. It is introduced as “the fault zone” but is actually made of multiple fault strands (principal slip zones) and a damage zone, and within the damage zone there are “faults and fractures”, so it is difficult sometimes to keep this all straight. What do the three boreholes actually intersect? It looks like Tests 1 and 2 are performed directly on one of the principal slip zones, whereas Test 3 is in the damage zone. Are both Tests 1 and 2 performed on a clay-bearing fault? This is of course important because clays have very different hydromechanical properties compared to fractured limestone. Is there a significant difference between the location of Tests 1 and 2? If so, this could explain some of the differences seen in Figure 4, for example the permeability fit in Test 3 may be related to the large overall permeability of the damage zone, whereas the less well-fit permeability model in Tests 1 and 2 could be the effect of clay gouge. But if the location of Tests 1 and 2 are the same, that does not explain the difference between these two.

We agree that more information about the fault zone architecture and the geology of the borehole intervals where injection tests are performed is needed. We have clarified this point by improving the Figure 1b, as well as in the text on lines 59-69 and 77-84 by adding details.

On lines 83-89, we now present details for each test interval:

Inside the injection chamber, the pressurized fault plane is located at the middle of the interval between two inflatable packers. Tests 1 and 2 are situated along a principal slip zone, while Test 3 is in a fractured damage zone. In Test 1, the rock matrix around the pressurized fault plane contains three subparallel bedding planes in the sealed borehole interval, while in Test 2, the rock matrix contains micro-fractures without bedding planes. In Test 3, the pressurized fault plane is surrounded by a fractured matrix with a fracture density higher than in Tests 1 and 2. In the three tests, the selected planes are not filled with clay material.

We think that some differences seen in the permeability and friction evolution in Figure 4 can be due to some differences in the borehole interval geology in and around the pressurized fault plane. Following the suggestion of the reviewer, we have now included additional text on lines 306-313:

Although some differences in the fault response observed between the tests can be due to the different levels of fracturing of the rock matrix that surrounds the pressurized fault plane, the exact mechanical process responsible for permeability increase or decrease after the sudden acceleration remains elusive. Based on geological observations of the borehole intervals before injection, they could reflect geometrical changes of contact areas, gouge content and void space that comprise the fault during deformation at low effective stress for Tests 1 and 2, whereas the hydromechanical response observed in Test 3 may be related to the overall permeability of the fractured damage zone surrounding the tested fault segment.

I'd like a bit more information on the measured permeability values. What exactly is meant by the permeability of the "rock matrix"? Intact limestone, or the damage zone which includes fragments of intact limestone and a network of fractures? What is meant by the permeability of the fault zone? Across-fault permeability? Along-fault permeability? Is this a fault with clay gouge?

The permeability of the rock matrix corresponds to the intact limestone. We have clarified by adding "*intact rock matrix*" (lines 97 and 132).

With our in-situ protocol, the fluid pressure is increased step-by-step in a borehole interval intersecting the fault plane. Thus, the permeability of the fault zone is an apparent permeability, integrating both the across- and along-fault components. We have clarified by adding "*apparent fault permeability*" (line 134).

I found the calculations and relation of permeability to slip to be somewhat convoluted. Both the permeability increase and slip activation are the result of dilatancy (which is of course the result of reduced effective normal stress). So I am not sure if it is really correct to say that the permeability is a function of the slip (i.e., they are both effects of the same process but one does not necessarily cause the other) (e.g. Line 18). For the calculated permeability shown in Fig 4, why calculate it from the slip, which is calculated from the aperture opening, and not directly from the aperture opening itself? I think it also would help if the main goal/result of the study is clearly stated, "co-evolution of friction and permeability is important" is not very clear. I guess the goal is to be able to calculate permeability and friction and/or frictional stability from a known pressure perturbation and measured dilatancy and/or slip? I suggest stating something like this directly.

We have clarified this point. The permeability change is due both to reducing effective normal stress and slip-induced dilation. We have added "*reducing effective normal stress*" (line 18), "*during increasing fluid pressure*" (lines 51-52). In the discussion (lines 326-328), we also added new sentences to directly state the main objective and result.

In Figure 3 (dots) and Figure 4 (orange line), the permeability change estimated from experimental data is directly calculated from the measured opening caused by increasing fluid pressure using equation 1. To clarify this point, we have added a sentence in the Methods (lines 390-392). Then, the modeled permeability (black line) is calculated from the opening associated with the slip induced by fluid pressure perturbation. In the model, slip is sensitive to the change in effective normal stress, and so slip increases upon an increase in the fluid pressure as observed experimentally (Fig. 3b, Fig. 4a,c,e). We now clarify this point on lines 173-174.

Care should be taken when referring to "seismic slip" vs "aseismic slip". All the seismicity is off-fault (or at least remote from the test location in the borehole. So really, all the measured slip here is actually aseismic, and the "seismic slip" is triggered on a different fault somewhere else. On a related note, on Lines 288-290 it is stated that "small values of slip-weakening distance" (I guess here the critical slip distance D_c is meant?) are suggested to be responsible for remotely triggered seismicity, but the RSF parameters and critical nucleation lengths obtained here are only relevant for the fault for which the slip was modeled. For remote seismicity, the RSF and elastic properties for that fault have to be evaluated.

We agree with this comment. We have changed to “critical slip distance” (lines 342-343) and mentioned that the values are estimated at the injection points.

We have also added a new sentence (lines 413-416) in the Methods to clearly mention that “*the frictional and elastic properties in the zones of seismicity observed at a distance of injection points are not evaluated in the present study. A future characterization of the frictional parameters within the whole fault zone may help for better defining the seismic potential of fault segments and fractures around the injection.*”

Care also needs to be taken with the term “evolution of friction”, I guess you mean here the friction time series as shown in Fig 4. But the friction parameters that determine fault stability (the rate- and-state friction parameters) are fixed for a particular modeled pressure step, and do not evolve, or co-evolve with permeability (e.g. Line 22).

We agree, this is the transient evolution of friction and permeability. We have added “transient” to clarify this point (lines 22, 189, 365). Hence, we have modified the title to state “*Transient evolution of permeability and friction in a slowly slipping fault activated by fluid pressurization*”.

Line 87: what samples were used in these experiments? Limestone? Clay gouge?

The laboratory experiments were performed on fault gouge collected from rock samples drilled from the fault zone, which we reactivated by fluid injection at a depth in the underground gallery (Cappa et al., *Sci. Adv.*, 2019). We have clarified this point in the text on lines 102-103.

Reference :

Cappa, F., Scuderi M.M., Collettini C., Guglielmi Y., Avouac J.P. Stabilization of fault slip by fluid injection in the laboratory and in situ, *Sci. Adv.*, 5(3), eaau4065, doi: 10.1126/sciadv.aau4065, (2019).

Line 107: so the seismicity occurs only in the damage zone, and not on any of the clay-lined principal slip zones? this makes sense based on what is known about clay friction in general but is worth mentioning

The seismicity indeed occurs in the fractured damage zone around the fault segment activated by the fluid pressurization. This point is mentioned in lines 121-124 with a reference to previous work on the seismicity analysis (see Duboeuf et al., *JGR*, 2017). However, this is not because of clay friction (as no clay is present in the tested fractures), but to the large nucleation lengths around the injection, as stated on lines 341-343.

Reference :

Duboeuf, L., De Barros, L., Cappa, F., Guglielmi, Y., Deschamps, A., Seguy, S. Aseismic motions drive a sparse seismicity during fluid injections into a fractured zone in a carbonate reservoir. *J Geophys. Res.*, 122, 8285-8304, doi: 10.1002/2017JB014535, (2017).

Lines 218-223: $\Delta\phi$ is described as a “dilation parameter dependent on slip velocity” but ϕ is conventionally porosity, so is $\Delta\phi$ just a porosity change? It seems simpler to describe it this way.

Indeed, in the initial equation proposed by Fang et al. (2017), $\Delta\phi$ is described as a “dilation parameter dependent on slip velocity”. To avoid confusion with the conventional porosity (ϕ), we have changed $\Delta\phi$ by Δv in equations 6 and 7, and on line 249.

Lines 264-267: it would be helpful to report the rate-and-state friction values from the previous study, so the reader can directly compare.

Thanks for this suggestion. We now give on lines 316-317 the rate-and-state friction values estimated from the previous laboratory experiments on fault gouge (Cappa et al., Sci. Adv., 2019).

Line 311: as noted above, they are both linked to dilatancy (aperture opening) and reduced effective normal stress

We have clarified by adding the following sentence “*both linked to effective normal stress and slip*” in lines 365-366.

- **Reviewer #2 (Remarks to the Author):**

In their manuscript “Co-evolution of permeability and friction in a slowly slipping fault activated by fluid pressurization”, Cappa et al. presented the results of three in-situ injection experiments where fluid pressure, fault displacement (normal and parallel) and slip velocity are measured. They then used a numerical model relating the hydraulic aperture (assumed to be equal to fault-normal displacement), permeability, slip, slip velocity, and rate-and-state friction to fit the observed permeability and fault slip data, finding a reasonably good fit. This study corroborates with some previous experimental studies they and others have done, showing the importance of permeability enhancement upon slip triggering. The results are very interesting to the seismology community, especially for those modeling injection-induced aseismic slip. I do have a few concerns regarding the data fit, and would appreciate if the authors could address the following comments.

We thank you for the helpful, constructive feedback, and your suggestions. We have tried to address all your questions.

Main comments

1. The fit to Test 1 vs. Tests 2 & 3 show some rather significant differences. The reason why Test 1 does not have as great fit is explained in the Discussion, but when it was first mentioned in Lines

242-246, the explanation was rather vague and general, perhaps adding some explanations there will make the reader more convinced, and then in the discussion, those points can be further developed. However, in concept, all tests involve similar set-ups, so whatever has not been taken into account in the model for one test also wasn't taken into account for the other tests. So why is there much less discrepancy for the other two tests?

We have added more explanations about the possible reason why Test 1 does not have as great fit in Lines 275-282, to help the reader before the Discussion on this specific point:

The exact process responsible for such a difference remains subtle. It could be related to additional hydromechanical effects or geometrical complexities ignored in the model we use. In such well-developed fault zone, it could also reflect heterogeneous fluid flow (i.e. channeling) over the pressurized fault segment or a fluid leakage from the pressurized fault into the surrounding connected fractures or bedding planes. As the injected fluid volume in Test 1 is much larger than for the other tests, it is likely that an extended network of faults is pressurized, leading to more complex hydromechanical response.

Indeed, although tests involve a similar experimental set-up and are close each other in the fault zone, the geology of the borehole interval is different (lines 77-84) and some subtle difference in the fluid flow (for instance, channeling versus uniform) could explain the discrepancy between tests.

2. I'm particularly interested in the iterative parameter search to produce the best fit. The values of d_c and K_s , for example, span over an order of magnitude across the different tests, which are very close to each other in space. Is it realistic to have them be so different for such similar test locations?

Although the test locations are close in the fault zone, geological and geometrical differences exist (lines 77-84). Indeed, there is difference in fracture density and rock elasticity (Jeanne et al., 2012) around the pressurized fault that could explained the different value of K_s . The difference in d_c could be due to different degree of roughness as observed in Jeanne et al., 2012. We now mention this in the caption of the Supplementary Table 1.

Reference :

Jeanne P., et al. (2012), Architectural characteristics and petrophysical properties evolution of a strike-slip fault zone in a fractured porous carbonate reservoir, J. Struct. Geol., 44, 93-109, doi: 10.1016/j.jsg.2012.08.016

3. Is there any consideration of using the mass balance equation to fit the fluid pressure evolution using the parameters found above? Maybe that will be a good check on whether they are not just (over)fitting the permeability profile, if they can also reasonably fit the pressure profile.

This is an interesting point. With the precise hydromechanical measurements at injections points reported in this paper, the behavior of the fault zone is quite complex and fully coupled hydromechanical modeling is needed to reproduce the pressure profile. In future work, we intend to develop additional modeling with new experiments including more monitoring boreholes in the fault and distributed sensors to better track the three-dimensional fluid pressure evolution over time. With the current data and models, we feel that the new discovery presented here is quite important and relevant to our study.

4. My concern with the hydraulic-aperture and permeability relation is that, if the fault has opened so much to produce the observed order of magnitude enhancement in permeability, then why is there no obvious dilatant suction effect seen in the pressure profile?

In the experimental protocol used in the study, the injection corresponds to a pressure-control mode to maintain a quasi-constant pressure at each hydraulic loading. In short, we impose the fluid pressure in the injection chamber between inflatable packers. The fluid pressure is a boundary condition and was increased step-by-step. We now clarify this point on lines 72-73. Consequently, fluid pressure drop during fault opening are limited at the injection. We think that additional measuring points in the fault around the injection would have helped to see potential dilatant suction effect. Additionally, a constant flow rate boundary condition could be more appropriate to detect this effect.

Minor comments

1. Equation (1) the cubic law is written as square

To avoid confusion, we have removed “cubic law” in line 138.

2. Line 152: do the constants beta and gamma have meanings?

Beta represents how much dilatancy is involved when the fault slips. Gamma is geometric constant. We have clarified in lines 171-172.

3. Line 178: could you explain why this assumption is valid for high initial permeability fault?

This assumption is valid for faults with sufficiently high permeability, such that the characteristic timescale of fluid diffusion is smaller than the timescale of deformation. In other words, the pressure is assumed to be drained at all times. We have clarified on lines 199-200.

4. Lines 182-183: do Tests 1 & 2 show induced seismicity? The slip rates as shown in supplementary Figure 2 are quite low for all cases and never reach seismic speeds?

Tests 1 and 2 show induced seismicity over the time window selected for the modeling. Figure 2D and 2E illustrates this seismicity which is located around the fault segment pressurized by fluid injection, which slips aseismically at slow velocity consistently with data presented in supplementary figure 2. We have added “(Fig. 2a and e)” on line 205.

5. Lines 262-263: could you expand on what is interesting about the permeability and friction evolving concurrently?

We have added the following sentence on lines 298-305:

Interestingly, the permeability and friction evolve concurrently under conditions of varying stress. This observation suggests a direct link between the transient evolution of permeability and friction presumably through rearrangement of fault surface asperities as a result of slip and opening or closure. The comparison between experiments shows two fault behaviors. In the first case, a rate-weakening behavior is associated with a fault closure after the pressure increase (Test 2), and in the second case, a rate-strengthening behavior is observed together with a fault opening (Test 3).

6. Figure 2: the green dots are labeled as distance of seismic events to injection, but where is the scale that shows how many far it is to injection?

The scale that shows how many far a seismic event is to injection is the right axis labelled “*Seismic events*”. As indicated in the legend mentioned in Figure 2D, this axis is used both for the cumulative number of seismic events and the distance of seismic events to injection in meter. For clarity, we have added the significance of each axis in the figure caption (lines 572-574).

- **Reviewer #3 (Remarks to the Author):**

A review of “Co-Evolution of Permeability and Friction in a Slowly Slipping Fault activated by Fluid Pressurization by Cappa F., Giglielmi Y, De Barros L.

This manuscript presents a large-scale in-situ experiment on permeability and friction evolution in a natural fault. The authors performed fluid injections in a km-scale fault zone in limestone in the LSBB lab (France). They measured fluid pressure, normal and shear displacements, and seismic activity. Permeability and friction are modelled from these data. This topic is highly relevant for unconventional reservoir geomechanics. The main result presented by the authors is that friction and permeability are dependent on both slip and slip velocity.

The paper is well written and the main messages contained in it are clearly delivered and well supported by experimental and modelling data. It definitely deserves to be published. However,

the work would better fit in a more specialized journal, where part of the interesting supp. mat. could be incorporated into the main text. I have only moderate comments.

We thank you for your constructive feedback and for recognizing the broad interest and applicability of this study and for placing the work into context. We have tried to address all your questions, and now included the suggested information in the main text and in additional supplementary figures.

Comments:

L14: in-situ (ita)

Changed to italic was done for all “*in situ*”.

L31: Specify the frictions laws and permeability laws derived from lab tests

Done in lines 32-34.

L41: Cite Zappone et al., 2021, solid earth?

Done as suggested.

L58-60, Fig 1: Add fault orientation, and movement on fig 1

Done as suggested on Figure 1B and in Supplementary Figure 1B.

L 66-72: Please give more details on injections protocols in supp. mat. For example, justify the injection depth, durations, volumes, max pressures ect...

Thanks for this suggestion. Details on the injection depth, test durations, injected fluid volumes and maximum pressure are given directly in the main text on lines 71-92. Other details can also be found in Duboeuf et al. (JGR, 2017) as mentioned in the text. Respectfully, we consider they are acceptable and sufficient information about the details on injection protocols.

L 72: Similarly, give the locations of the seismic sensors (accelero, and geophones) in fig 1. r supp. mat.

Thank you for this suggestion. We added a new panel with the locations of seismic sensors in the Supplementary Figure 1.

L90: Careful here! The lab experiments were performed on gouge material. The natural fault is rough at different scales and without gouge material (L 62-63). It is well known that the dependences of friction to slip, and slip rate are strongly related to fault structure (with (a-b) increasing with slip-velocity in gouge material, and decreasing on bare surfaces, Noel et al., (2019)). Moreover, the static friction coefficient of gouge material is different than the one for bare surfaces. Note also that gouge material has almost no cohesion.

Thanks for suggesting that. We have clarified by adding “*on gouge material collected from rock samples drilled from the fault zone*” on lines 102-103.

Fig 2: Add a marker at the beginning of the different deformation stages in Fig. 2: elastic deformation stage, fault opening, slip acceleration, creep

Done as suggested. See new Figure 2.

L113: Is it the permeability or the transmissivity that is measured? How is the initial fault effective thickness measured?

We used equation 1 to estimate the permeability change from the fault-normal displacement directly measured at the injection point. Please, see the section “*Estimate of the fault permeability change from the temporal evolution of fault-normal displacement*” in the Methods (lines 379-392).

The initial fault effective aperture was estimated in previous studies from the analysis of a series of hydraulic injection based on pulse and step-rate tests of different magnitudes and durations (Guglielmi et al, Science, 2015; Jeanne et al., 2012) (lines 147-148).

References :

Guglielmi, Y., Cappa, F., Avouac, J.-P., Henry, P., Elsworth, D. Seismicity triggered by fluid injections induced aseismic slip. *Science*, **348**(6240), 1224–1226, doi:10.1126/science.aab0476, (2015).

Jeanne, P., Guglielmi, Y., Lamarche, J., Cappa, F., Marié, L. Architectural characteristics and petrophysical properties evolution of a strike-slip fault zone in a fractured porous carbonate reservoir. *J Struct. Geol.*, **44**, 93–109, doi: 10.1016/j.jsg.2012.08.016, (2012).

L120: How is normal fault aperture linked to hydraulic aperture?

We assume that the change in hydraulic aperture is equivalent to the change in normal displacement, consistently with previous experimental works of the hydromechanical response of permeable fractures in *in situ* conditions (Cornet et al., IJRM, 2013; Cappa et al., IJRM, 2006).

This point is mentioned in the Methods (please, see the section “*Estimate of the fault permeability change from the temporal evolution of fault-normal displacement*”) (lines 379-392).

References :

- Cappa F., Guglielmi Y., Rutqvist J., Tsang C-F., Thoraval A. Hydromechanical modelling of pulse tests that measure fluid pressure and fracture normal displacement at the Coaraze Laboratory site, France, *Int J Rock Mech Min Sci*, **43**, 1062-1082, doi: 10.1016/j.ijrmms.2006.03.006, (2006).
- Cornet F.H., Li L., Hulin J.P., Ippolito I., Kurowski P. The hydromechanical behaviour of a fracture: an in situ experimental case study, *Int J Rock Mech Min Sci*, **40**, 1257-1270, doi: 10.1016/S1365-1609(03)00120-5, (2003).

L 125: Give the hypothesis behind equation 1: A single fracture is reactivated (i.e. interplay with different fault segments is not taken into account), incompressible fluid of uniform density, steady state pressure ect... Moreover, please acknowledge that this equation does not consider the complexity of the aperture field due to contact roughness.

Thank you for the suggestion. We added a sentence as suggested on lines 142-143.

L132: It is not clear to me how fault parallel displacement is calculated. Is it calculated at the end of each step?

The fault-parallel displacement is directly measured at the injection point during the experiment using the SIMFIP borehole probe (lines 88-92). To generate the figure 3, the data are selected in the time series of fault-parallel displacement as the selection of fluid pressure and fault-normal displacement data presented in Supplementary Figures 4-5, that is a stabilized value before activation of a pressure step.

L149: Related to previous comment, what are the hypothesis behind equation 2? If I am not wrong the equation is derived from the CNS model developed for granular material. How does complex fault roughness, geometry, multiple segments influence the link between dilatancy and slip? (with probable complex fluid velocity field, slip, slip rate in all directions). Here, slip is measured only at the injection point.

This is correct, equation 2 was originally derived from the CNS model (Niemeijer and Spiers, JGR, 2007) developed for granular material, and then successfully applied to reproduce both laboratory measurements (slip and dilation) on fault gouge and in situ measurements (slip and dilation) on natural faults (Van den Ende et al., Solid Earth, 2017). Interestingly, the model fits well the measured permeability into the three in situ experiments presented here. Although the model geometry simplifies the natural conditions of the experiment (roughness, geometry, segmentation, fluid flow, etc.), our numerical results are consistent with the observed fault hydromechanical response. Reproducing this permeability evolution is the main objective of the modelling presented here. We feel that this new discovery is quite important and relevant to our study. We also want to emphasize that such hydromechanical data collected in situ directly in fault zones are rare and

generally difficult to acquire. Thus, the aim of our study is to be able to understand the complex processes arising from the interaction between fluid pressure, fault opening and slip.

References :

Niemeijer, A. R. and Spiers, C. J.: A Microphysical Model for Strong Velocity Weakening in Phyllosilicate- Bearing Fault Gouges, *J. Geophys. Res.*, 112, B10405, doi: 10.1029/2007JB005008 (2007).

Van den Ende, M.P.A, Scuderi M.M., Cappa F., Ampuero J.P. Extracting microphysical fault friction parameters from laboratory and field injection experiments, *Solid Earth*, 11, 2245-2256, doi: 10.5194/se-11-2245-2020, (2020).

L173: Cite Noel et al., 2019, JGR?

Thanks for this suggestion of reference. Cited as suggested.

L175: in-situ

Changed to italic.

L176: How have the 3 sequences been selected?

The three sequences have been selected at the higher-pressure steps with similar initial pressures of 3.67 (Test 1), 3.83 (Test 2) and 3.4 MPa (Test 3) before the step-increase, and a slip that is well established on the fault. In addition, in order to have the clearest possible stepped-pressure perturbation, we have intentionally chosen short sequences (tens of seconds) during the course of injection when the fault motions are well developed, and the magnitude of slip and normal opening is comparable. These two points are presented on lines 202-209.

L217: Cite Samuelson et al., 2009?

Thanks for this suggestion of reference. Cited as suggested.

L217: Remind the reader why friction and, even more importantly, permeability depend on slip and slip velocity in Fang's experiments (shear enhanced dilatation, change in contact distribution? ect...), and justify why this is also true (or not) in the in-situ experiments described here. Note that Acosta et al., have shown different results, such that shear reduces dilatancy. Dilatancy strongly depends on initial roughness and slip.

As suggested, we have added details about this point in lines 240-248.

Friction and permeability of faults are known to change during slip due to the rearrangement and destruction of asperities. Permeability may increase due to shear induced dilation⁶ or decrease as a result of gouge formation³⁴ or irreversible changes in surface roughness³⁵. During slip, frictional strength is affected by the state of the contact area between the displacing surfaces^{4,32-33}. Previous field experiments on faults have also demonstrated that the contribution of slip on the evolution of permeability and friction is central^{11,13,16,36}. This gives a degree of confidence about the applicability of slip- and slip velocity-dependent models inferred from laboratory data for the analysis of in situ data.

Based on our previous works published in 2015 (Guglielmi et al., Science, 2015) and 2018 (Cappa et al., 2018), we showed from in situ experiments that the contribution of fault slip on the permeability changes is central. The most recent laboratory experiments of Fang et al. (JGR, 2017) and Ishibashi et al. (WRR, 2018) are on the same line. This gives a degree of confidence about the applicability of such slip- and slip velocity-dependent models inferred from laboratory data for the analysis of in situ data.

References:

- Cappa, F., Guglielmi, Y., Nussbaum, C., Birkholzer, J. On the relationship between fault permeability increases, induced stress perturbation, and the growth of aseismic slip during fluid injection. *Geophys. Res. Lett.*, **45**, 11,012–11,020, doi: 10.1029/2018GL080233, (2018).
- Fang, Y., Elsworth, Wang D.C., Ishibashi T., Fitts J.P. Frictional stability-permeability relationships for fractures in shales, *J. Geophys. Res.*, **122**, 1760–1776, doi:10.1002/2016JB013435, (2017).
- Ishibashi, T., Elsworth, D., Fang, Y., Riviere, J., Madara, B., Asanuma, H., Watanabe, N., Marone, C. Friction-stability-permeability evolution of a fracture in granite. *Water Resour. Res.*, **54**, doi: 10.1029/2018WR022598, (2018).
- Guglielmi, Y., Cappa, F., Avouac, J.-P., Henry, P., Elsworth, D. Seismicity triggered by fluid injections induced aseismic slip. *Science*, **348**(6240), 1224–1226, doi:10.1126/science.aab0476, (2015).

L 217: Is it possible that mechanisms such as flux-driven unclogging of the fracture possibly link frictional evolution to permeability change in the presented experiment?

The mechanism of flux-driven unclogging with particle mobilization is a plausible scenario but difficult to study in the present in situ experiment. We mentioned this mechanism on line 359 as possible within the fault.

L232/figure 4: I would be curious to see more of those pressure steps fitted by the model.

In the model, the pressure step is imposed a loading condition. Thus, the pressure data are not fitted by the model, but correspond to an input parameter. The model is used to reproduce the fault displacements, and to infer evolution of permeability and friction.

L262: Please explain why some fault portions do experience dilation why others see compaction.

This is a good point. The exact mechanical process responsible for fault dilation or contraction remains elusive. They could reflect geometrical changes of contact areas and void space that comprise the fault during deformation at low effective stress. We have added new explanations on lines 299-309.

Interestingly, the permeability and friction evolve concurrently under conditions of varying stress. This observation suggests a direct link between the transient change in permeability and friction presumably through rearrangement of fault surface asperities as a result of slip and opening or closure. The comparison between experiments shows two fault behaviors. In the first case, a rate-weakening behavior is associated with a fault closure after the pressure increase (Test 2), and in the second case, a rate-strengthening behavior is observed together with a fault opening (Test 3).

Although some differences in the fault response observed between the tests can be due to the different levels of fracturing of the rock matrix that surrounds the pressurized fault plane, the exact mechanical process responsible for permeability increase or decrease after the sudden acceleration remains elusive. Based on geological observations of the borehole intervals before injection, they could reflect geometrical changes of contact areas, gouge content and void space that comprise the fault during deformation at low effective stress for Tests 1 and 2, whereas the hydromechanical response observed in Test 3 may be related to the overall permeability of the fractured damage zone surrounding the tested fault segment.

L285: Remind the reader that seismicity is localized far from the injection point (where measurements are performed).

Done as suggested. We have added “which showed that the seismicity is localized far from the injection point where the hydromechanical measurements are performed” on lines 338-339.

L287: Give L_c for test 3.

Frictional analysis of Test 3 indicates a rate-strengthening frictional behavior with a positive (a-b), which is a suitable condition for aseismic slip. Thus, we provide L_c only for the Tests 1 and 2 with a rate-weakening frictional behavior which favors potential seismic slip.

L330: Provide a zoom of several steps in supp mat. I would like to see the steady-state fluid pressure conditions, that seems not completely achieved in fig 4 of supp mat.

Thanks for this suggestion. We added an additional Supplementary Figure 5 with close-up views of different pressure steps over the pressure range from 0 to 3 MPa, and from 3 MPa to the maximum injected pressure. This new figure better illustrates each pressure steps. Ideally, one

would be able to clearly define the start and end of each pressure step. However, it is also important to mention that during fault activation experiments in the field, it is difficult to reach the ideal stability conditions generally obtained in laboratory experiments under a well-controlled environment. In nature, boundary conditions are not controlled, and it is possible to reach a near steady-state pressure. Thus, we have also added “*near steady-state fluid pressure conditions*” in the Methods (line 384).

**REVIEWERS' COMMENTS**

Reviewer #1 (Remarks to the Author):

I think the authors have done a good job of addressing the reviewer comments, and find that the paper is ready for
publication. Thorough checking for grammar would not hurt, though.

Reviewer #2 (Remarks to the Author):

The author has addressed all of my comments, which I find satisfactory. I recommend this article for publication in
Nature Communications.

Reviewer #3 (Remarks to the Author):

This is a substantially revised manuscript. The author has done a very nice job of improving the science
communicated in the paper. I think the paper can be published as it is, after the authors have a chance to consider
the2 following very minor comments:

L103: natural gouge material, or remoulded material?

L217: justify the use of the aging law and not of the slip law